# Investigation of the Fatigue Stress of Orthotropic Steel Decks Based on an Arch Bridge with the Application of the Arlequin Method

**DOI:** 10.3390/ma14247653

**Published:** 2021-12-12

**Authors:** Cheng Cheng, Xu Xie, Wentao Yu

**Affiliations:** 1College of Civil Engineering and Architecture, Zhejiang University, Hangzhou 310058, China; 11812059@zju.edu.cn; 2The Architectural Design & Research Institute of Zhejiang University Co., Ltd., Hangzhou 310058, China; yvette_lrk@chd.edu.cn

**Keywords:** orthotropic steel deck, premature fatigue cracks, finite element analysis, Arlequin framework, engineering guidance

## Abstract

Orthotropic steel decks are widely used in the construction of steel bridges. Although there are many fatigue-evaluation methods stipulated by codes, unexpected fatigue cracks are still detected in some bridges. To justify whether the local finite element model commonly used in fatigue investigations on orthotropic decks can correctly instruct engineering practices, the Arlequin framework is applied in this paper to determine the full fatigue stress under traffic loads. The convergence on and validity of this application for orthotropic decks are checked. Results show that the Arlequin model for deck-fatigue analysis established in this paper tends to be an efficient method for complete fatigue stress acquisition, whereby the vulnerable sites of orthotropic steel decks under traffic loads are defined. Vehicles near the flexible components, such as hangers or cables, can have adverse effects on the fatigue durability of decks. Additionally, the total number of vehicles and their arrangement concentration also affect fatigue performance. Complex traffic conditions cannot be fully loaded in local models. Regardless of the gross bridge mechanics and deck deformation, the fatigue stress range is underestimated by about 30–40%. Such a difference in fatigue assessment seems to explain the premature cracks observed in orthotropic steel decks.

## 1. Introduction

Orthotropic decks with closed trough stiffeners have been widely applied in the construction of steel bridges because of their favorable characteristics, such as large carrying capacity, low dead weight, short construction period, etc. This application is particularly common for long-span cable systems or special-shaped bridge systems. Nevertheless, it is well known that orthotropic steel decks are liable to fatigue once placed in service. Under heavy live loads, fatigue damage appearing in vulnerable areas will continually be amplified and eventually lead to structural failure. Fatigue cracks have been reported in many practical bridges [1,2,3,4,5], and similar failures were also observed in a series of experimental investigations [6,7,8]. To provide effective guidance for orthotropic deck design, great efforts have been made to study mechanical fatigue behaviors and their evaluation methods. So far, the perception that welded structures’ fatigue life is mainly affected by stress amplitude Δσ has been generally accepted by scholars. Furthermore, fatigue-evaluation criteria based on this concept for steel structures were also correspondingly developed with different specifications [9,10,11,12,13,14].

However, there are still many more fatigue cracks than expected observed in bridges designed by these specifications [15,16,17]. The most probable reason for this phenomenon is the underestimation of the fatigue stress caused by traffic loads, which leads to the occurrence of cracks in the early stages of service and reduction of structures’ service life [18,19]. Because of the great complexity of mechanical properties, dividing the orthotropic deck structure into three sub-systems and analyzing it layer-by-layer is a common approach for simplification. On the basis of this division, many structural calculation methods have been developed, such as orthotropic plate theory [20,21], lattice girder theory [22], difference method [23] and finite element (FE) simulation. Among them, FE simulation has been adopted by general researchers due to its convenience for use in local stress obtention. There are a few studies on the proportion of these three fatigue stress components. For instance, Tong et al. compared these three components according to a load test and simple simulations of a steel girder bridge [24]. Results showed that the orthotropic deck’s stress condition is mainly aroused by the local effect of the wheel load; in other words, the third fatigue stress is dominant. Following this viewpoint, researchers would like to adopt a refined local model to replace the cumbersome full bridge model for the sake of calculation costs. This local approach has been applied in recent investigations [25,26,27,28,29,30]. However, there are two defects with this approach. First, the conclusion that the local effect is dominant was only derived from a simple girder bridge form; its generality is equivocal. Furthermore, for bridges with unclear mechanical transmissions, such as long-span systems and cable-stayed systems, the multi-vehicle effect is not included. Additionally, restricted by computing power, previous studies on orthotropic decks were based on rough FE analysis. The credibility of stress acquisition is unwarrantable. So far, similar research in this field has been limited.

The aim of this paper is to discuss the validity of the traditional local approach and the multi-vehicle cooperative effect in orthotropic decks’ fatigue stress calculation. Given that the final failure of structures is related to the global response under external factors as well as the local damage accumulation, global/local structural analysis is currently considered to be an efficient tool for analyses of structural details. To pursue full fatigue stress, the Arlequin theory proposed by Professor Ben Dhia [31], which has greater flexibility in the modeling process and stronger acceptance in the theoretical establishment [32,33,34], is combined with the FE analysis in this paper. Taking a special-shaped tied arch bridge as an example, full bridge FE models, including weld details, are established with the application of Arlequin technology. The accuracy of the Arlequin algorithm for orthotropic steel decks is verified first. Then, according to the results of the local model and the global model, the commonly used FE method is evaluated. Finally, based on traffic measurements on domestic urban bridges, the differences of fatigue stress aroused by a single vehicle and traffic loads are also discussed. This paper provides some references for the fatigue evaluation of orthotropic steel bridge decks.

## 2. Description of the Bridge

A through-tied steel arch bridge in Jiaxing City, China, was used as the research objective of the current study. As shown in Figure 1, two asymmetrical arch ribs are set along the transverse direction: the rise-span ratio of the big arch rib is 1:3, its arc rise is 39.33 m and its inclined angle is 15°; the small one is characterized by a 1:5 rise-span ratio and 23.6 m of arch height; the out-inclination is 20°. Each rib has outside and inside hangers. The main girder of this bridge is constructed as a variable box cross-section with a single span of 118 m, which is made of Q345C steel [35]. The mechanical properties are shown in Table 1, where *E* indicates the elastic modulus; *σ*_y_ and *σ*_u_ denote the yield strength and ultimate strength, respectively; and *A* represents the elongation ratio. Its top plates adopt the orthotropic steel deck form with closed U-ribs, fabricated by cold bending of 8 mm steel plates. All the U-ribs pass through the 30 mm diaphragm set every 4 m. The bottom plates of the main girder are composed of orthotropic plates with L-shaped stiffeners, which are made of 200-type flat-bulb steel. In the carriageway area, the thickness of the top plates and bottom plates is 18 mm and 14 mm, respectively. Details of the trough-to-deck weld joint are shown in Figure 1d.

## 3. Arlequin Method

As depicted in Figure 2, the overall model is divided into two regions—Ω_1_ and Ω_2_—and the overlapping area of these two regions is defined as coupling region *S*. According to analysis requirements, different mesh sizes and element types can be assigned in Ω_1_ and Ω_2_. The primary purpose of the Arlequin method is to establish transition elements in coupling region *S* to conduct global/local analysis, with the usage of the energy partition function and reliable coupling operators between Ω_1_ and Ω_2_. The following is a brief explanation of the stiffness matrix of the transition elements.

Suppose that *u_i_* denotes the displacement tensor of region Ω*_i_*. ***λ*** is the Lagrange multiplier within the coupling area, and Γ*_u_*, *f*, *ε*(*u*) and *σ*(*u*) represent the displacement constraints, load tensor, strain tensor and stress tensor of boundary ∂Ω*_i_*, respectively. Then, the possible displacement fields in each region can be expressed as:(1)$$\left\{ \begin{array}{l} W_{i}=\left\{\begin{array}{cc} u_{i}\in H^{1}\left(\Omega_{i}\right); & u_{i}=0\;\mathrm{on}\;\Gamma_{u} \end{array}\right\}\\ W_{\lambda }=H^{1}\left(S_{\lambda }\right) \end{array}\right.$$
where the operator *H*^1^ is the scalar product defined in Sobolev space. Assuming $W_{hi}\subset W_{i}$ and $W_{h\lambda }\subset W_{\lambda }$ denote the FE discrete spacing of regions Ω_*i*_ and *S_λ_*, respectively, the Arlequin algorithm for this model can be interpreted as the solution to the following saddle point equation:(2)$$\underset{\left(u_{h1},u_{h2}\right)\in W_{h1}\times W_{h2}}{\min}\left\{\underset{\lambda_{h}\in W_{h\lambda }}{\max}\left[E_{1}\left(u_{h1}\right)+E_{2}\left(u_{h2}\right)+C\left(\lambda_{h},u_{h1}-u_{h2}\right)\right]\right\}$$
where $E_{i}\left(u_{hi}\right)=\frac{1}{2}\int_{\Omega_{i}}\alpha_{i}\sigma \left(u_{hi}\right):\varepsilon \left(u_{hi}\right)d\Omega -\int_{\Omega i}\beta_{i}f\cdot u_{hi}d\Omega$ is the weighted virtual work in region Ω_*i*_, and the weight function *α_i_* and *β_i_* of strain energy and work satisfy the following relationship:(3)$$\{\begin{array}{cc} \alpha_{i}=\beta_{i}=1 & \mathrm{in}\;\Omega_{i}\backslash S_{\lambda }\\ \alpha_{1}+\alpha_{2}=\beta_{1}+\beta_{2}=1 & \mathrm{in}\;S_{\lambda } \end{array}$$

The coupling operator *C* in Equation (2) is defined as:(4)$$C\left(\lambda ,u\right)=\int_{S_{\lambda }}\left(\lambda \times u+l^{2}\varepsilon \left(\lambda \right):\varepsilon \left(u\right)\right)d\Omega$$
where *l* is the scaling parameter. If applying $\varphi_{1}^{i}$, $\varphi_{2}^{j}$ and $\varphi_{\lambda }^{k}$ as the primary functions in spacing ***W***_h*i*_ and ***W***_h*λ*_, $\hat{u_{i}}$ and $\hat{\lambda }$ as the coordinates of ***u***_h*i*_ and ***λ***_h_ along these primary functions, a variational solution for Equation (4) can be obtained:(5)$$\left[\begin{array}{ccc} K_{1} & 0 & C_{\lambda 1}^{T}\\ 0 & K_{2} & -C_{\lambda 2}^{T}\\ C_{\lambda 1} & -C_{\lambda 2} & 0 \end{array}\right]\left[\begin{array}{c} \hat{u_{1}}\\ \hat{u_{2}}\\ \hat{\lambda } \end{array}\right]=\left[\begin{array}{c} F_{1}\\ F_{2}\\ 0 \end{array}\right]$$
in which ${\left(K_{i}\right)}_{jk}=\int_{\Omega_{i}}\alpha_{i}\varepsilon \left(\varphi_{i}^{j}\right):\varepsilon \left(\varphi_{i}^{k}\right)d\Omega$ and ${\left(F_{i}\right)}_{j}=\int_{\Omega_{i}}\beta_{i}f\cdot \varphi_{i}^{j}d\Omega$ indicate the weighted stiffness matrix and load column vector for the coupled area, respectively, and ${\left(C_{\lambda i}\right)}_{jk}=\int_{S_{\lambda }}\left(\varphi_{\lambda }^{j}\times \varphi_{\lambda }^{j}+l^{2}\varepsilon \left(\varphi_{\lambda }^{j}\right):\varepsilon \left(\varphi_{i}^{k}\right)\right)d\Omega$ is the coupling matrix. The first part of the left side of Equation (5) is the stiffness matrix of the transition elements.

## 4. Finite Element Models

In this section, the establishment and rationality of the global model and the local model are elaborated first.

### 4.1. Global Model

Figure 3 shows the global FE model, with the application of the Arlequin framework. Geometrical contents such as arch ribs, hangers, box girder, diaphragms and stiffeners, etc., are included. The boundary conditions are six circular bearings set at the two ends of the bridge, and the longitudinal, transverse and torsional DOF, respectively, are constrained according to actual placement. The traffic load applied on the deck panel is composed of several wheel loads. The setting of each wheel load refers to the twin wheel load specified in [36]: the contact area is 0.2 m × 0.6 m and 0.2 m for the longitudinal direction and 0.6 m for the transverse direction; the axel load is 120 KN and 0.5 MPa for every wheel-pressure magnitude. For each global FE model, a refined solid area of 2.1 m × 1.7 m will be embedded under a wheel action area, on which the geometrical details of the weld joints are modeled. Hangers are simulated by the two-node linear hybrid truss element T3D2H; the refined solid part is simulated by the eight-node reduced integration solid element C3D8R, which is one-quarter fine meshed plate thickness; the rest of the steel plates are simulated by the four-node reduced integration shell element S4R, whose mesh size is 0.3 m. Solid elements located along the edge of the local refined zone and its adjacent shell elements are operated by the coupling element subroutine integrated by the Arlequin algorithm on the ABAQUS-6.14 platform. The basis for this mesh size and local refined area selection is explained in Section 4.2.

### 4.2. Local Model

The local model used in this paper replaced the simplified approaches applied in previous fatigue studies, as shown in Figure 4. The modeling content is a 3 m × 3 m solid orthotropic deck section, containing five U-ribs and one diaphragm. The load condition is a single-wheel load located at the center of the top plate. Displacements on the nodes along the border of the local model are constrained.

Due to the limitations of the element characteristics, it is generally accepted that at least four layers of solid elements on the plate thickness can meet the bending stress in shell elements for thin-walled structures [37]. The determination of mesh size is carried out by a mesh sensitivity check according to three local models with different mesh arrangements, as shown in Figure 4. Stresses on the bottom surface of the deck panel calculated by these three models are compared, as depicted in Figure 5. Results show that, although the stress values of the stress peak points near the weld joint increase slightly as mesh fineness increases, the stress distribution almost keeps constant in a general view. Given the cost and accuracy, the mesh standard for solid modeling is determined as a one-quarter plate thickness partition. This conclusion is also applied in local refined regions of global models.

The check for the convergence of the Arlequin algorithm is also performed, as shown in Figure 6. Four Arlequin models integrated with different solid areas are established under a 3 m × 3 m bridge deck range, in which the solid area increases gradually from type A to type D. Together with the full solid version, a comparison is made of the stress distributions of these five models. Results show that with the enlargement of the solid region, the Arlequin solution tends to approach a full solid solution. Minimal gaps are captured between C and D-type Arlequin models and the full solid model, of which the error is within 2%, indicating that these two kinds of solid volume layouts can already eliminate the multi-scale effects caused by the load area of the wheel load. For the sake of cost analysis, the C-type arrangement was chosen in this paper. The global model shown in Figure 3 can be regarded as the result of extending the shell part of the local model to a full bridge range.

## 5. Fatigue Stress Analysis of Orthotropic Steel Deck

According to the comparison of these two types of numerical work, the validity of the conventional local model can be evaluated. Although the calculation criteria for fatigue life vary in different specifications, fatigue life and fatigue stress are of linear correspondence. Thus, the stress change rate *δ* is defined as the evaluation index in this paper, which can be expressed as:(6)$$\delta =\frac{\sigma_{g}-\sigma_{l}}{\sigma_{l}}\times 100\%$$
where *σ_l_* and *σ_g_* are the stress range derived from the local model and the global model, respectively. According to the degree of stress concentration derived from the stress nephogram in subsequent simulations, A, B, C and D nodes are selected as the most vulnerable sites for stress comparison, as shown in Figure 7. The stress nephogram is depicted in the next section. For vulnerable points B–D, the stress range is the nominal stress by direct calculation; for point A located at the root of the weld toe, the stress range is the hot spot stress obtained by the linear extrapolation formula recommended by the International Institute of Welding (IIW) [38]:(7)$$\sigma_{\mathrm{hs}}=1.67\sigma_{0.4t}-0.67\sigma_{1.0t}$$
where *σ*_0.4t_ and *σ*_1.0t_ are the nodal stresses at reference points 0.4 times and 1.0 times the plate thickness away from the weld toe, respectively, as illustrated in Figure 7.

### 5.1. Fatigue Stresses of the Local and Global Models

The fatigue evaluation criteria for the steel bridge specified in the design codes have no strict provisions for the acquisition of fatigue stress, which is the main parameter of these methods. For the reduction in analytical and experimental costs, taking only a local part of the bridge deck as the research object is common practice, while the fatigue stress obtained by this local analytical method is incomplete as a result of the ignorance of gross down-warping and deck deformation. Moreover, due to the limitations of the model size, certain fatigue vehicles often cannot be fully loaded. Additionally, vehicles can drive in any lane, and the location between the center of the road line and the cable side is random. Nevertheless, the local model can only contain a stereotyped loading and boundary mode. Such complex vehicle conditions cannot be taken into account. In this section, following the specifications of [36], the Type III fatigue vehicle is taken as the standard vehicle load, and four representative locations are selected on the deck panels of the global model for loading: Midspan–Road Centre (MC), Midspan–Cable Side (MS), Quarter Span–Road Centre (QC) and Quarter Span–Cable Side (QS), as depicted in Figure 8. Each rectangle in this figure represents a fatigue vehicle, under which a wheel is selected to be put on refined solid regions. The local model is loaded by a single wheel.

A numerical result for the global model is displayed in Figure 9, in which the stress concentrations correspond to the four vulnerable points (Figure 7). Table 2 presents the specific stress value and stress change rate *δ*. From these data, it is obvious that fatigue stress calculated at the vulnerable sites in the bridge deck is significantly amplified by the gross bridge system, while the intervention of different load positions is relatively small. For any monitoring points, fatigue stress simulated by the global model is more than 20% greater than the result of the local model, indicating that the practice of local simulation is somewhat radical under the current fatigue design code. The vulnerable point D is an exception as its stress change rates are up to 50%. The phenomenon of such tremendous stress fluctuation is caused by the smaller stress value of point D because of the constancy of stress differences between these four points. Furthermore, the stress responses of groups MC and QC, and groups MS and QS, are of great consistency. The fatigue stress of the latter group is larger than the former. This indicates that for bridges with complex mechanical systems, the local wheel load actioned near the flexible structures, such as hangers or cables, would have adverse effects on the fatigue performance of the bridge deck. In terms of the spatial effect of the vehicle load, the transverse influence is greater than that of the longitudinal direction.

### 5.2. Fatigue Stresses Provoked by Other Vehicle Loads

Actual traffic flows consist of multiple vehicles, and the location and number in each traffic lane are random. The fatigue stress of the orthotropic deck is not only directly aroused by the vehicle loaded on the top of the vulnerable sites, but also indirectly affected by the vehicles in other positions due to the deck deformation. Because of limited geometrical content, the local model for orthotropic deck fatigue analysis cannot satisfy the variable traffic flow conditions. The synergy of other vehicles is neglected.

Detecting all the fatigue performances under each traffic condition is time-consuming and laborious work. Given that the main purpose of this paper is to explore whether the local FE models can correctly guide engineering design, the comparisons are only made on some typical domestic traffic statuses. The design of the traffic load is determined by the number as well as the spacing of vehicles in each carriageway. To ensure the fatigue stress aroused by the designed traffic is of considerable referential importance, the headway refers to the statistical data of domestic urban bridges’ traffic conditions as 15 m [39,40], and the vehicle number was arranged based on the design load [36]. Following the above restrictions, four traffic flow cases with different vehicle distributions were designed, as shown in Figure 10. Each rectangle in the figure represents the standard fatigue car, among which the brown one represents the car acting on top of the coupled regions, and the rest represent the other vehicles loaded on the bridge at the same time. All four Arlequin models have the same location as the solid location of the MC model.

Table 3 shows the numerical results, in which *δ* and *δ*_1_ are calculated based on the stress of the MC model and the local model, respectively. As can be observed, compared with single-vehicle loading, fatigue stress aroused by traffic loads increases about 10%. For some vulnerable sites, the stress change rate can be up to 14%. This indicates that the practice of single-fatigue-vehicle loading stipulated in the design code can roughly reflect real traffic conditions. From traffic flow case 1 to case 4, the outright load applied on the bridge is constant, but the concentration of vehicle distribution around the refined solid region is gradually weakened. Correspondingly, the stress change rate *δ* of each column in Table 2 has a gradual decline from top to bottom. This means that the fatigue performance of the bridge deck is not only related to the total number of vehicles, but its scattered status is also a key factor. The stress change rate *δ*_1_ reflects the effects of the three sub-systems on fatigue stress calculations; its impact is about 30–40% for each traffic condition. As far as the arch bridge analyzed in this paper, the local approach is not suitable for fatigue evaluation because of the huge stress difference. The practice of obtaining fatigue stress through local analysis is not available for all bridge structures. This may help to explain the premature cracks observed in bridges under service.

## 6. Conclusions

The fatigue evaluation of orthotropic steel decks with the application of a multi-scale algorithm is carried out in this paper. Taking a special-shaped steel arch bridge as an example, the differences of fatigue stress simulation between the local model and the global model and fatigue performance under traffic loads are discussed. The following conclusions are obtained:

1. Multi-scale FE models of bridge decks established under the Arlequin framework in this paper can take into account the weld details of the vulnerable parts, as well as the structural mechanics of the overall bridge, providing an effective analytical method for fatigue evaluation of orthotropic steel decks.

2. Numerical results under the Arlequin framework indicate four stress concentration positions of orthotropic decks A–D, which are located near the weld toe–deck intersection, weld hole–trough adjacence and diaphragm arc, respectively. Among them, vulnerable sites A–C have higher stress values, which are susceptible to fatigue cracks.

3. The fatigue stress range, regardless of the gross bridge mechanics and deck deformation, is significantly underestimated, with 32%, 35% and 43% declines for vulnerable sites A, B and C, respectively, indicating that the local FE simulations applied in previous studies are not reliable for all bridge types. Such great influence on fatigue stress evaluation seems to explain the premature cracks in orthotropic steel decks.

The fatigue stress range and its cycling number are two fundamental factors in the fatigue life of welded structures. This paper mainly discusses the stress simulations of fatigue assessment. As for the conjoint effect of these two factors, further study is required.

## Figures and Tables

**Figure 1 materials-14-07653-f001:**
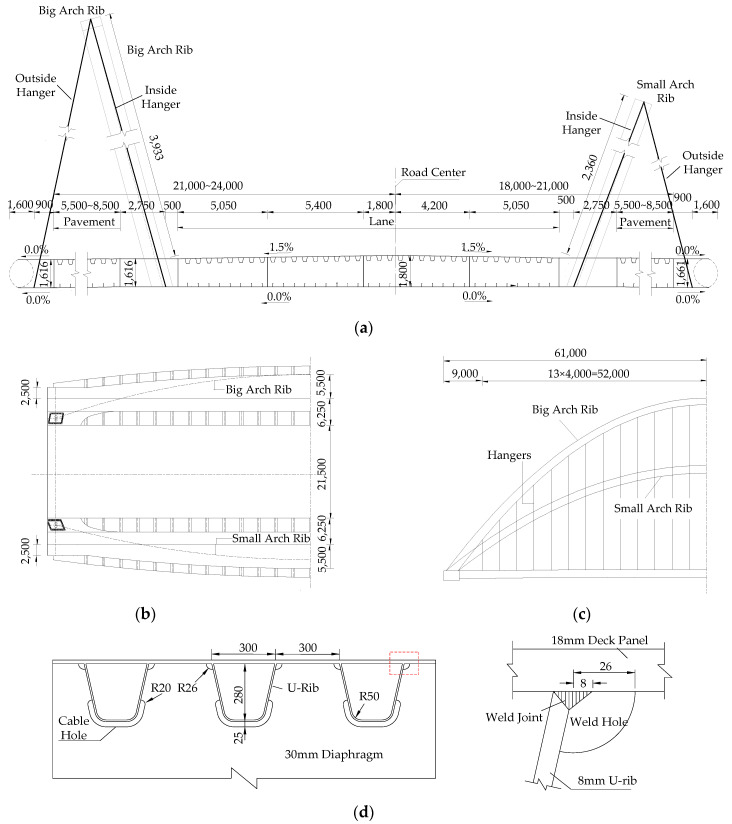
Schematic diagram of the arch bridge: (**a**) deck cross-section at the center span; (**b**) plan view; (**c**) elevation view; (**d**) structural details of orthotropic deck (unit: mm).

**Figure 2 materials-14-07653-f002:**
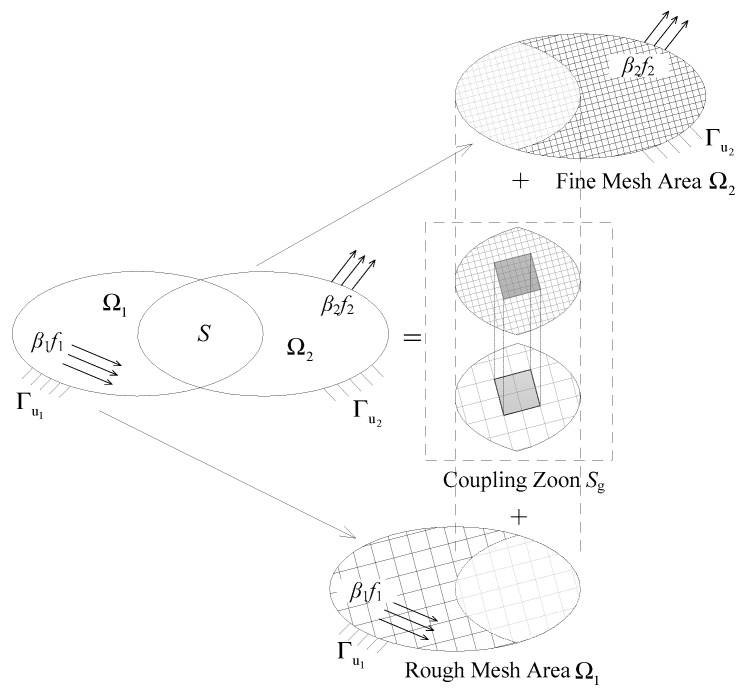
Regional division for Arlequin method.

**Figure 3 materials-14-07653-f003:**
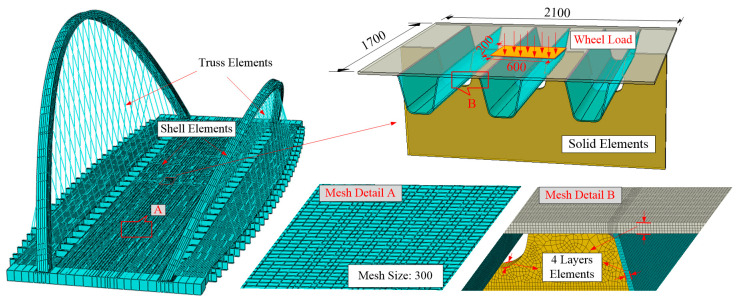
Sketch of the global model (unit: mm).

**Figure 4 materials-14-07653-f004:**
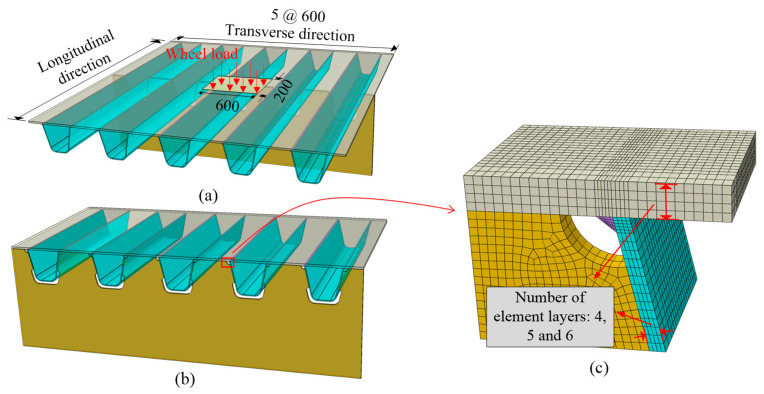
Schematic diagram of the local model (unit: mm): (**a**) full model; (**b**) half model; (**c**) mesh details.

**Figure 5 materials-14-07653-f005:**
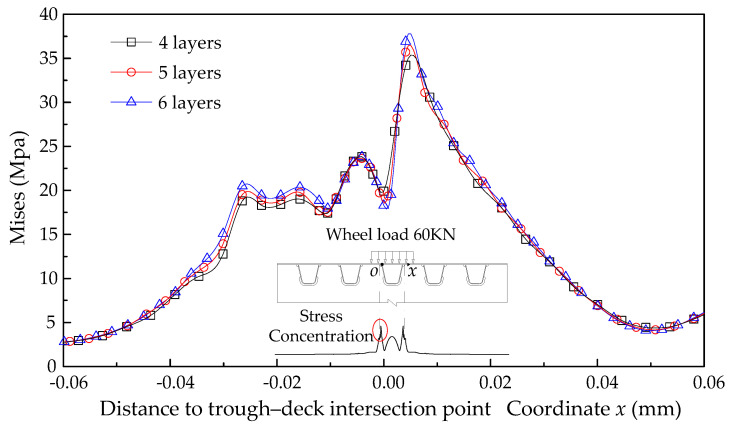
Mesh sensitivity check of the local model.

**Figure 6 materials-14-07653-f006:**
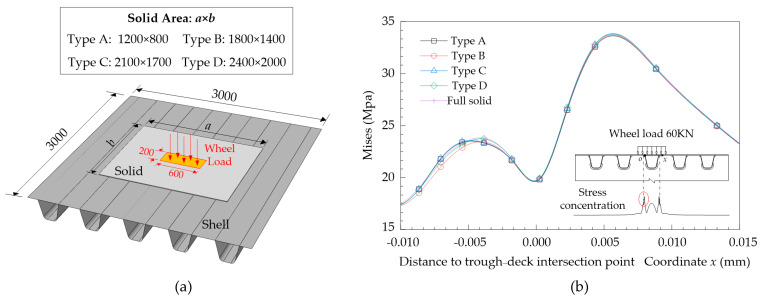
Convergence verification: (**a**) influence of different solid area; (**b**) validation model (unit: mm).

**Figure 7 materials-14-07653-f007:**
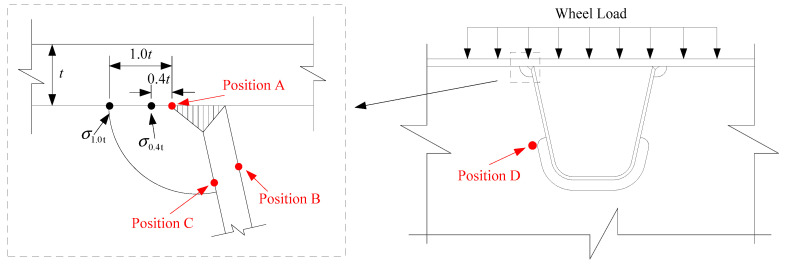
Vulnerable position of orthotropic deck.

**Figure 8 materials-14-07653-f008:**
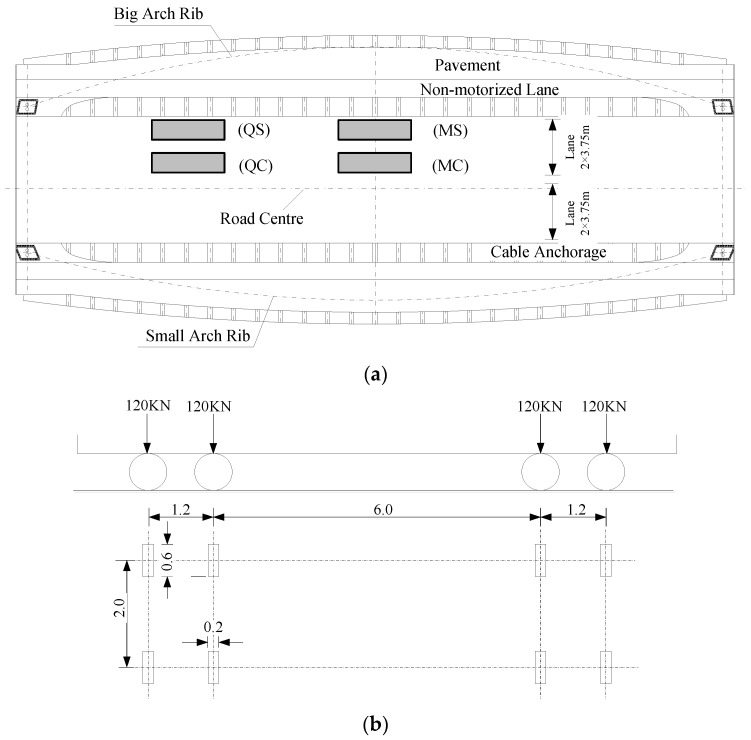
Load case for the global model: (**a**) load location on the deck panel; (**b**) fatigue vehicle load [35].

**Figure 9 materials-14-07653-f009:**
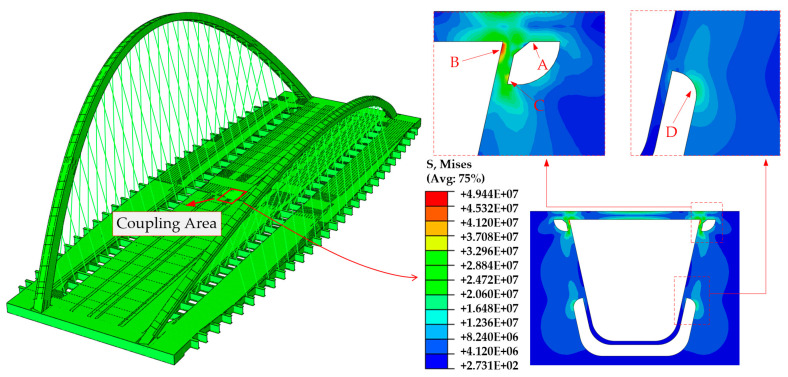
Numerical result of the Arlequin model (MC model).

**Figure 10 materials-14-07653-f010:**
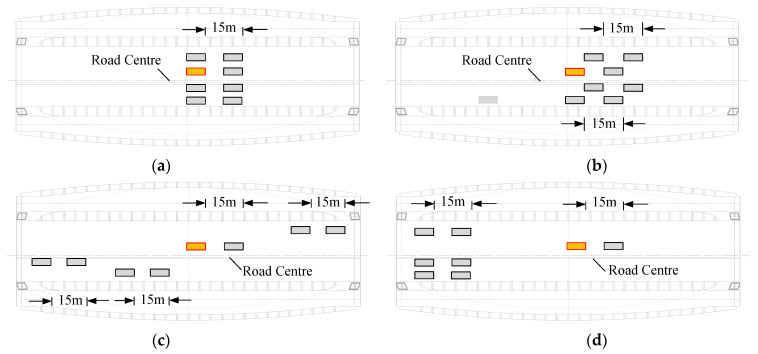
Four traffic-flow cases: (**a**) case 1; (**b**) case 2; (**c**) case 3; (**d**) case 4.

**Table 1 materials-14-07653-t001:** Mechanical properties of Q345C steel.

*E* (MPa)	*σ*_y_ (MPa)	*σ*_u_ (MPa)	A (%)
198,221	351.10	508.57	40.60

**Table 2 materials-14-07653-t002:** Stress results of the global model and the local model (unit: MPa).

Model	Position A		Position B		Position C		Position D	
	Stress Range	*δ*	Stress Range	*δ*	Stress Range	*δ*	Stress Range	*δ*
Local Model	23.0	-	51.2	-	44.4	-	16.0	-
MC	27.8	20.87%	62.3	21.68%	55.7	25.45%	24.3	51.87%
MS	28.3	23.04%	62.9	22.85%	56.6	27.48%	25.4	58.75%
QC	27.5	19.57%	61.7	20.51%	55.2	24.32%	23.2	45.00%
QS	28.4	23.48%	62.2	21.48%	56.3	26.80%	24.8	55.00%

**Table 3 materials-14-07653-t003:** Stress results of different traffic flows (unit: MPa).

**Traffic Flow Case**	**Position A**			**Position B**		
	**Stress Range**	** *δ* **	** *δ* _1_ **	**Stress Range**	** *δ* **	** *δ* _1_ **
Local Model	23.0	-	-	51.2	-	-
MC	27.8	-	-	62.3	-	-
Case 1	30.3	8.99%	31.74%	68.9	10.59%	34.57%
Case 2	30.2	8.63%	31.30%	68.6	10.11%	33.98%
Case 3	29.6	6.47%	28.70%	65.7	5.46%	28.32%
Case 4	29.5	6.12%	28.26%	64.9	4.17%	26.76%
**Traffic Flow Case**	**Position C**			**Position D**		
	**Stress Range**	** *δ* **	** *δ* _1_ **	**Stress Range**	** *δ* **	** *δ* _1_ **
Local Model	44.4	-	-	16.0	-	-
MC	55.7	-	-	24.3	-	-
Case 1	63.3	13.64%	42.57%	26.4	8.64%	65.00%
Case 2	62.7	12.57%	41.22%	25.9	6.58%	61.88%
Case 3	59.3	6.46%	33.56%	23.5	−3.29%	46.88%
Case 4	58.1	4.31%	30.86%	24.1	0.82%	50.63%

## Data Availability

Not applicable.

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
