# Peer review of "Investigation of the Fatigue Stress of Orthotropic Steel Decks Based on an Arch Bridge with the Application of the Arlequin Method"

_materials, 2021, doi:10.3390/ma14247653_

Round 1

Reviewer 1 Report

The paper can be published after the corrections have been made. Please send the paper again after the authors make corrections.

Author Response

Please see the attachment. The corrections in the revised manuscript are address in red.

Reviewer 2 Report

The authors of the submitted paper deal with an analysis of stress in bridge construction. They focused on the investigation of fatigue stress of an orthotropic steel deck on an arch bridge by computer simulation by use of the Arlequin method. They present results on a selected model and discuss the validity of the presented method of the calculation.

Using the computer modelling of many engineering tasks can be very helpful. The manuscript is well written but I suggest a minor revision, as follows:

1) It seems that the solved model was validated only numerically. In this case, the shell authors explain in more detail why they did not perform verification also using experimental methods.

2) In the manuscript, Figure 8 is missing. There is probably an error in the numbering of figures.

3) Line 207: It is not clear to me what the abbreviation IIW means.

4) The format of References shall be checked. The guidelines for authors have to be adhered to precisely.

Author Response

(The authors gave the same response as above.)
